# Analysis of the Sustainability of Long-Term Detraining Caused by COVID-19 Lockdown: Impact on the Maximal Aerobic Speed of Under-16 Soccer Players

Ana Filipa Silva [1,2,3], Filipe Manuel Clemente [1,2,4], Georgian Badicu [5], Daniele Zangla [6,*], Rui Silva [1,2], Gianpiero Greco [7], Halil Ibrahim Ceylan [8], João Alves [1], Francesco Fischetti [7,†] and Stefania Cataldi [7,†]

1    Escola Superior Desporto e Lazer, Instituto Politécnico de Viana do Castelo, Rua Escola Industrial e Comercial de Nun'Álvares, 4900-347 Viana do Castelo, Portugal; anafilsilva@gmail.com (A.F.S.); filipe.clemente5@gmail.com (F.M.C.); rui.s@ipvc.pt (R.S.); joao.carlos.alves176@gmail.com (J.A.)

2    Research Center in Sports Performance, Recreation, Innovation and Technology (SPRINT), 4960-320 Melgaço, Portugal

3    The Research Centre in Sports Sciences, Health Sciences and Human Development (CIDESD), 5001-801 Vila Real, Portugal

4    Instituto de Telecomunicações, Delegação da Covilhã, 1049-001 Lisboa, Portugal

5    Department of Physical Education and Special Motricity, University Transilvania of Brasov, 500068 Brasov, Romania; georgian.badicu@unitbv.ro

6    Department of Psychology, Educational Science and Human Movement, University of Palermo, 90144 Palermo, Italy

7    Department of Basic Medical Sciences, Neuroscience and Sense Organs, University of Study of Bari, 70124 Bari, Italy; gianpiero.greco@uniba.it (G.G.); francesco.fischetti@uniba.it (F.F.); stefania.cataldi@uniba.it (S.C.)

8    Physical Education and Sports Teaching Department, Kazim Karabekir Faculty of Education, Ataturk University, 25240 Erzurum, Turkey; halil.ibrahimceylan60@gmail.com

\*    Correspondence: daniele.zangla@unipa.it

†    These authors share last authorship.

**Abstract:** This retrospective cohort study aimed to analyze the effect of a 9-week detraining caused by COVID-19 lock-down on the maximal aerobic speed (MAS) of youth soccer players. The study included twenty-two under-16 male players competing at the national league level (15.4 ± 0.7 years old) who were analyzed pre and post the detraining period. The MAS was estimated using the Bronco's test. Moreover, the self-regulated physical exercise performed by the players during the lockdown was monitored using a questionnaire. Considering the training volume per week, the players were grouped into those working more (>180 min/week) and those working less (<180 min/week) during the lockdown. Within-group changes revealed no significant differences ($p = 0.122$; d = 0.381) in the group that self-trained less than 180 min/week started (pre-lockdown) at 3.97 ± 0.29 m/s and ended (post-lockdown) at 3.85 ± 0.34 m/s, corresponding to a decrease of 3.02%. For those who self-trained more than 180 min/week during the lockdown, they started with 4.33 ± 0.28 m/s and decreased by 1.62% for 4.26 ± 0.28 m/s ($p = 0.319$; d = 0.250). The team as a whole (pooled data, all players included) decreased 2.27% from 4.15 ± 0.34 to 4.06 ± 0.37 m/s ($p = 0.077$; d = 0.321). Between group analysis revealed that the group of players that trained more than 180 min/weeks were significantly better than players working below 180 min/week in both pre-lockdown moment (+9.01%; $p = 0.007$; d = 1.263) and post-lockdown moment (10.6%; $p = 0.006$; d = 1.323). Home-based training can be a good strategy to mitigate the detraining effects caused by a lockdown.

**Keywords:** football; physical fitness; sports training; athletic performance; pandemic; youth; aerobic capacity

## 1. Introduction

Training principles are important to adjust the stimulus to the player's levels [1]. Reversibility is one of the five basic training principles from which the term "detraining" can be viewed as an integrated part of the reversibility principle [2]. Briefly, reversibility is based on the concept that a reduction in training leads to a partial or complete reversal of previous physiological adaptations [3]. The detraining concept is based on the first above-mentioned premise of reversibility principle, which stands for the partial or complete loss of the adaptations as a consequence of training cessation [2,4]. Normally, detraining is classified as short term (<4 weeks of training reduction/cessation) and long-term (>4 weeks of training reduction/cessation) [5]. Although coaches purposely partially reduce training as a taper strategy, if well-planned, it does not cause detraining effects. However, training cessation due to injury, illness or other unpredictable factors can have significant detrimental effects on the different biological human systems [6].

Considering the macrocycle of a soccer season, players are exposed to approximately 10 months of continuous high-demanding training and competition, divided by specific time-frame periods such as the pre-competition, competition and transition [7]. It is during the transition period (off-season), that a partial or complete training cessation occurs, resulting in long-term (>4 weeks) detraining [4]. Despite some athletes may take part in non-periodized sports activities beyond soccer specificities, a loss of their sport-specific adaptations occurs and ultimately impact player performance and their injury resilience levels to cope with competition demands [8]. Indeed, transition periods can be used as an opportunity for coaches to prepare the athletes for the high-volume and frequency of training typical from the pre-competition physical demands [7].

The information about detraining effects is now important because COVID-19 brought challenges regarding prolonged inactivity or low activity caused by a governmental lockdown [8]. In fact, the presence of the COVID-19 pandemic during the last two years made the entire training process in football very difficult, as the imposed restrictions and lockdowns resulted in long-term detraining effects [9]. To better understand the effects that COVID-19 lockdown has on a player's physical fitness, only a few studies have been published [9–12]. However, from the available research on this topic, only two studies were conducted on youth soccer players [9,11].

These studies have demonstrated inconsistent findings regarding the overall selected physical fitness changes after the COVID-19 lockdown on youth soccer players [9,11]. Indeed, body fat, change of direction, speed, and vertical jump performance were significantly affected by long-term detraining due to the COVID-19 lockdown [9]. Furthermore, the most negatively affected physical capacity on youth players was the aerobic capacity [9]. On the other hand, a recent study conducted on 29 youth soccer players revealed significant improvements in the change of direction, speed, and vertical jump performance after a COVID-19 lockdown in which players had a limited physical activity level [11]. The authors suggested that the observed improvements could be associated with the maturation process [11].

Considering the aerobic and intermittent nature of soccer, in which athletes have to perform repeated short bursts at high intensities during both training and matches, a high aerobic power is beneficial to better cope with the game demands [13]. The maximal aerobic speed (MAS: lowest running speed at which maximum oxygen uptake occurs) [14] measure has the potential to indicate the capacity of an athlete to cope with the specific training demands of his sport [15]. Indeed, low MAS values are related with lower ability to cope with invasion-based field sports demands [13]. Given that, MAS is an important measure to consider when unexpected breaks, such as the COVID-19 lockdown, occur, as this measure may significantly decrease after a long-term detraining period.

To the best of the authors' knowledge, no study conducted on youth soccer players have examined the effects of a long-term detraining period on MAS performance due to the COVID-19 lockdown. Coaches, practitioners and athletes may potentially benefit from the understanding of such long-term detraining effects on aerobic power to better plan

and operationalize the return to training of their athletes. Given all the above-mentioned reasons, the aim of the present study is to analyze the effect of a 9-week detraining caused by the COVID-19 lock-down on the MAS of youth soccer players.

## 2. Materials and Methods

### 2.1. Experimental Approach to the Problem

The present study followed an observational, analytical cohort design. A convenience sample was analyzed for their aerobic fitness on 1 September 2020 and 2 November 2020 (62 days in between). Between the assessment periods, there were no guided field training sessions, as a result of the cessation of the club's intervention period. During the interruption period (9 weeks), the athletes were free to perform home-based self-guided activities (mentioning the type and regularity of these to the observers). Before 1 September 2020 (first assessment), the players had two weeks of pre-season training sessions.

### 2.2. Participants

Twenty-two under-16 male soccer players of a youth team that competes at national league level (15.4 ± 0.7 years of age; 174.2 ± 6.7 cm; 64.5 ± 7.3 kg; 6.6 ± 1, 2 years of experience in soccer training) participated in the present study. Players were enrolled from the start of the study and no drop-out or missing data were registered. During the period, no COVID-19 infection, illness, or injury was reported by the players. Based on the median amount of time of self-exercise reported, we have organized the samples into two groups: (i) those performing less than 180 min exercise/week ($n = 11$), and (ii) those performing more than 180 min exercise/week ($n = 11$). The 180 min/week were selected based on the median of reports of players about home-based training during the period of lockdown. The eligibility criteria were defined as follows: (i) athletes were assessed at the beginning and at the end of the cohort, (ii) the athletes did not suffer serious or disabling injury or illness during the cohort. All participants were informed about all the procedures. After being informed, they signed a free and informed consent form that mentioned the possibility of withdrawing at any time during the study. The study followed the ethical standards for research on human subjects set out in the Declaration of Helsinki.

### 2.3. Procedures

Participants were assessed for the following measures, sequentially: (i) anthropometry, and (ii) the Bronco test. Anthropometric assessment was performed in an isolated space with a mild temperature (23 °C), and with the athletes performing the test after food intake due to the test being performed in the late afternoon. After the anthropometric assessment, all athletes were exposed to a standardized warm-up consisting of 5 min of low-intensity running (self-paced), 5 min of dynamic stretching (focused on lower limbs), and 3 min of rest until performing the bronco test on artificial grass.

#### 2.3.1. Body Composition

Two measurements were carried out, consisting of body mass (kilograms) and stature (centimeters), in a changing room at the stadium. The instruments used were the following: (i) tape measure, and (ii) digital scale. For measuring stature, a tape was glued to the wall. Each athlete had to remove their shoes and lean against the wall. Each athlete was measured for one time, and the stature (centimeters) was taken.

#### 2.3.2. Physical Fitness Measures

Physical fitness assessments were taken at two different moments (pre-post). On 1 September 2020, the first assessment was carried out at the stadium synthetic turf and its time of completion was at 7:30 p.m. On 2 November 2020, the second assessment was carried out at the training center synthetic turf (similar to the turf presented in the stadium) at 7:30 p.m. Both assessments were monitored by a coach from the technical team. The

assessments took place on the same day of the week after 3 days of recovery considering the last training session.

### 2.3.3. The Bronco Test

All athletes performed the Bronco test (also known as the 1200-m shuttle run test). Athletes had to do the test in the shortest time possible, starting at the start line, running to the next line which was 20 m away. They then returned to the start line and performed another forward run to another line which was 40 m away. They returned to the start line once again before making a final run forward to a line which was 60 m away and finish the test by returning to the start line. The athletes had to repeat the procedure a total of five times, thus completing the 1200 m. Athletes were previously familiarized with the test, thus increasing the accuracy of their pace during the test. The total time, in seconds, required to complete the test was recorded. MAS was then calculated by dividing the distance of 1200 m by the recorded time. However, the time recorded for each player was corrected according to the following equation: MAS = 1200/(time in seconds—20.3) [16]. The 20.3 s represents the corrective approach for each 180° change of direction and it is used as a correction for light body mass [16].

### 2.3.4. Physical Activity Questionnaire

A questionnaire was applied every week using the Google Forms platform. All athletes had to answer the following questions: (i) full name; (ii) if you ever trained during the break derived from confinement, how many times a week you performed structured home-based exercise?; and (iii) how long each training session took? Based on the median time of self-exercise performed per week (median of the 9 weeks considered), the sample was organized into two groups: (i) players performing in average (of the nine weeks) less than 180 min exercise/week ($n$ = 11); and (ii) players performing on average (of the nine weeks) more than 180 min exercise/week ($n$ = 11).

### 2.3.5. Statistical Analysis

The descriptive statistics are presented in the form of mean and standard-deviation. Normality and homogeneity of the sample were tested using the Shapiro–Wilk and Levene's test, respectively. A mixed ANOVA tested the interaction between time (pre vs. post) and group (below and above 180 exercise minutes/week). The Bonferroni test was used as post-hoc test, and the Cohen's D effect size was also used pairwise comparisons. The statistical analysis was performed using the IBM SPSS statistics software (v.28.0.0.0, IBM, New York, NY, USA) for a $p < 0.05$.

## 3. Results

The Shapiro–Wilk revealed normality of the data in the pre-test below 180 min ($p$ = 0.393), pre-test above 180 min ($p$ = 0.212), post-test below 180 min ($p$ = 0.669) and post-test above 180 min ($p$ = 0.450). The Levene's test also confirmed the homogeneity in both pre-test ($p$ = 0.931) and post-test ($p$ = 0.561).

Mixed ANOVA (time*group) revealed non-significant interactions (F = 0.176; $p$ = 0.680; $\eta_p^2$ = 0.009). Within-group changes (Figure 1) revealed no significant differences ($p$ = 0.122; d = 0.381) in the group that self-trained less than 180 min/week started (pre-lockdown) at 3.97 ± 0.29 m/s and ended (post-lockdown) with 3.85 ± 0.34 m/s, corresponding to a decrease of 3.02%. For those who self-trained more than 180 min/week during the lockdown, they started with 4.33 ± 0.28 m/s and decreased by 1.62% for 4.26 ± 0.28 m/s ($p$ = 0.319; d = 0.250). The team as a whole (pooled data, all players included) decreased 2.27% from 4.15 ± 0.34 to 4.06 ± 0.37 m/s ($p$ = 0.077; d = 0.321). Between group analysis revealed that the group of players that trained more than 180 min/weeks were significantly better than players working below 180 min/week in both pre-lockdown moment (+9.01%; $p$ = 0.007; d = 1.263) and post-lockdown moment (10.6%; $p$ = 0.006; d = 1.323).

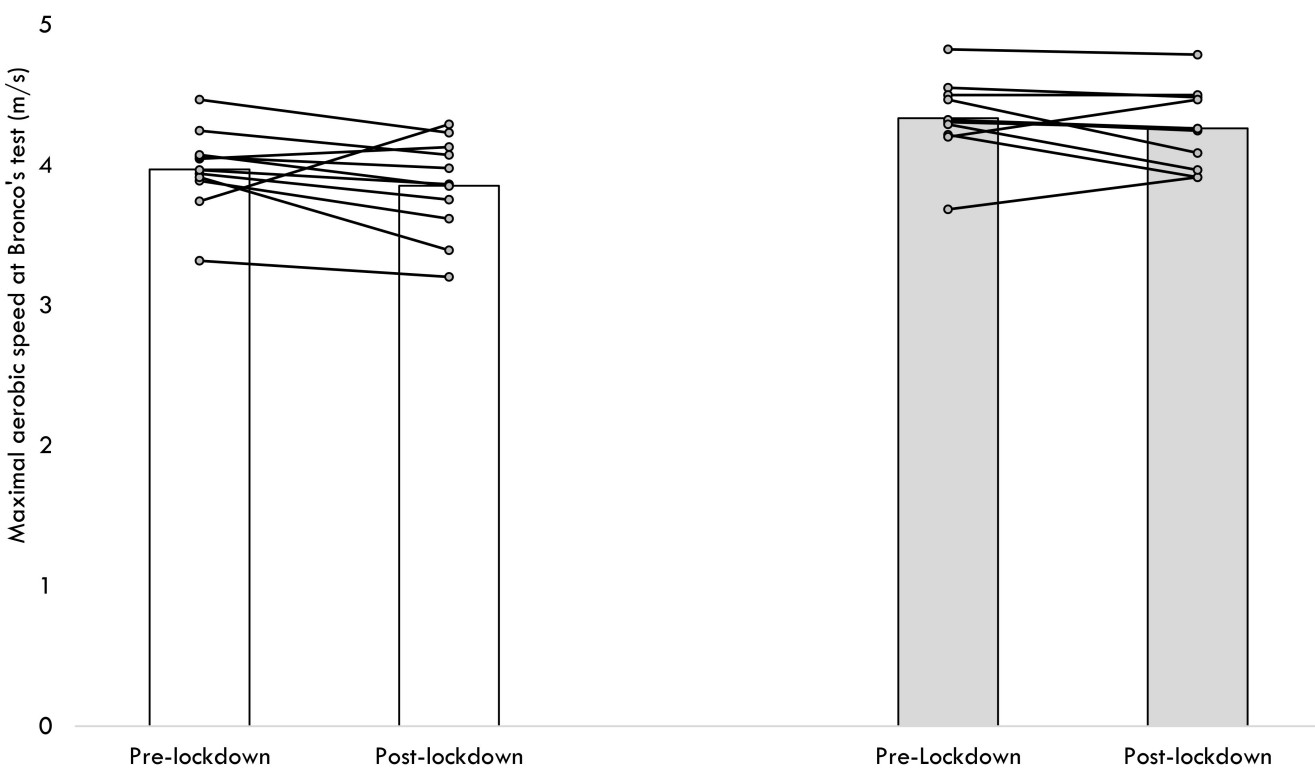

**Figure 1.** Pre–post lockdown variations of maximal aerobic speed at Bronco's test considering players who self-trained less than 180 min/week (white bars) and those self-trained more than 180 min/week (grey bars).

## 4. Discussion

As far as we know, no studies performed on youth soccer players have examined the effects of a long-term detraining period on the MAS performance due to the COVID-19 lockdown. Therefore, the aim of the present study is to analyze the effect of a 9-week detraining caused by the COVID-19 lockdown on the MAS of youth soccer players. The main result of this study was that detraining caused by the COVID-19 lockdown, regardless of groups (although both groups were physically active to a certain level), led to statistically significant deteriorations in MAS performance at Bronco's test.

Aerobic capacity is one of the most important indicators of physical fitness in soccer [17]. Time–motion analyzes show that adolescent elite young cover players (13–18 years old) cover a distance of approximately 6.5–9.0 km during the match, of which approximately 670–970 m is high-intensity activity, and 190–670 m is sprint distance [18]. Based on this information, soccer players need to have an advanced level of aerobic fitness in order to maintain repetitive high-intensity movements, accelerate the recovery process, and, in short, to cope with the physical and physiological demands of matches [19,20]. Detraining during the transition period caused significant performance impairments in a number of physiological and performance measures such as VO2max, time to exhaustion, repeated-sprint ability and Yo-Yo test intermittent-running performance [2,7,8,21]. Moreover, previous studies showed that a structured home-based exercise program during the COVID-19 lockdown was effective in maintaining or improving physical fitness measures such as countermovement jump performance, an indicator of lower extremity explosive strength in elite handball players [22], and youth soccer players [23]. In another study, it was stated that after a 32-week detraining period, caused by the COVID-19 pan-

demic lockdown, significant improvements were observed in neuromuscular parameters (vertical jump height, agility, and linear sprints) of young soccer players, depending on maturation-related adaptations [11].

The present study revealed that MAS did not change significantly in groups who self-trained less than 180 min or more per week during the COVID-19 lockdown. Additionally, it was observed that there were no significant reductions in MAS considering the team as a whole. These results suggest that the home-based exercise performed by both groups during this period was partially sufficient, and that the detraining caused by the COVID-19 lockdown on the MAS was possibly mitigated. Pucsok et al. [23] found that the endurance capacity of young soccer players was moderately deteriorated after completing a 13-week home-based training routine during the COVID-19 lockdown. In additon, recent studies stated that cardiorespiratory endurance or VO2max, determined by the maximum mean velocity reached in a multistage 20-metre shuttle run test, was decreased in elite handball players [24], youth soccer players [4,25], female soccer players [26], and professional soccer players [22] due to non-specific and insufficient stimuli caused by a home-based exercise program during the COVID-19 lockdown. Furthermore, contrary to neuromuscular parameters (countermovement jump, change of direction ability, and 30 m sprint performance), a recent study was conducted by Alvurdu et al. [9] on young soccer players aged 15–18 which noted that the highest performance loss was observed in aerobic capacity measured by the YYIRTL-1 test after a long-term detraining period of 15 weeks due to the COVID-19 lockdown. Similarly, in another study, young soccer players performed aerobic exercise twice a week and strength exercise for the development of upper and lower extremity muscles twice a week during the 2-month period of strict home confinement. Despite the exercise program in this process, the Yo-Yo test running distance and maximal running speed performance of the players decreased significantly after COVID-19 compared to pre-quarantine [27]. Additionally, Font et al. [22] reported that the aerobic capacity, measured by multistage 20-metre shuttle run test of top-level professional handball players, decreased during the COVID-19 lockdown. The same researchers further stated that the losses in aerobic power could be due to the fact that the training volume of home-based exercise programs was approximately 40% lower compared to the endurance training performed during the regular competitive season, and the training specificity was insufficient. Lastly, Mujika and Padilla [4] noted that when physical training is significantly reduced or stopped for more than 4 weeks, i.e., long-term detraining, athletes who train at a high level had a 6 to 20% reduction in VO2max. Contrary to the Mujika and Padilla study [4], a 2.27% decline in aerobic capacity was observed in the present study.

Considering the studies above, we suggest that despite the implementation of certain home-based exercise programs during the COVID-19 lockdown, these programs fail to provide an adequate stimulus to maintain aerobic performance, just as in our study, and long-term detraining induced decreases in aerobic fitness. The decrease in aerobic fitness can be caused by possible decreases in VO2max as a result of a significant decrease in training stimulus. Possibly, some justifications can be associated with decreased blood volume (together with a decrease in hemoglobin content), cardiac dimensions, number of mitochondria in cells, and respiratory efficiency, resulting in lower stroke volume and cardiac output despite increased heart rates. In addition, it has been suggested that reductions in capillarization, arterial-venous oxygen difference, skeletal muscle oxidative enzyme activities, and lactate threshold also contribute significantly to long-term losses in VO2max [4].

A previous study, which did not support the findings of our study, and was conduct ed on Italian Serie A League soccer players, was observed a significant improvement in aerobic fitness following the COVID-19 lockdown, and home-based training during lockdown was reported to be effective for improving aerobic fitness, although it did not allow players to maintain their power levels during competitive periods [28]. Likewise, existing evidence from the recent study of 20 elite professional soccer players, which does not overlap with the results of the present study, demonstrated that no significant changes were observed



in VO2max of the Yo-Yo test comparing before and after long-term (40 days) detraining due to the COVID-19 related restrictions and quarantine [20]. Moreover, the maintenance of aerobic capacity during the quarantine, despite long-term detraining, was associated with the content and intensity of the multi-dimensional soccer training program, including aerobic, resistance, balance, coordination, and specific power motor abilities with short running actions, accelerations, and decelerations, applied to the players. According to this, it was emphasized that multi-dimensional aerobic training performed at 65–75% of the maximum heart rate (3 times a week, 30 min) ensured the protection of aerobic capacity during the 40-day quarantine period, and might be a preservative health intensity [20]. Another study remarked that reduced training strategies could be used to delay the onset of long-term cardiorespiratory, metabolic, muscular, and hormonal detraining or to minimize losses in detraining in aerobic performance. Accordingly, maintaining training intensity might be the key factor for maintaining the physiological and performance adaptations caused by training, while training volume could be removed by 40 to 60% if training frequency is maintained. On the other hand, reductions in training frequency should be more modest (more than 20–30% in athletes and up to 50% in less trained people) [4,29].

The present study has some limitations. First, the sample size is small and was selected for convenience. Since no a priori sample size calculation was performed, any inference should be considered with caution. It may be difficult to generalize the present results. A second limitation is that the training intensity was not monitored. Third, in soccer, aerobic power or endurance capacity differs according to playing positions due to the different tactical roles in the matches during the COVID-19 Lockdown [30]. In our study, the effect of detraining on aerobic capacity was not shown in terms of different playing positions. Fourth, considering the effect of maturation on aerobic performance in young players [19,31], the maturation was not calculated in our study. The last limitation may be that the nutritional status of the athletes during the study was not followed or that the food consumption records are not recorded. Finally, no control group without exercising (total training cessation) was observed, which does not allow analyze the real impact of home-based self-exercise. In future studies, it is recommended to carry out more comprehensive studies investigating the relationship between detraining and performance-related parameters in different sports branches, taking into account the maturation, playing positions, gender factors, and training intensity.

## 5. Conclusions

The present study showed that the MAS performance of the groups that self-trained less than 180 min or more per week during the COVID-19 lockdown did not change significantly. However, it was observed that there was a 2.27% decrease in MAS at Bronco's test, regardless of groups. These results demonstrate that the self-regulated exercise performed by the players at home during the COVID-19 lockdown was insufficient in preserving and retaining cardiorespiratory fitness in elite youth soccer players. Lastly, our study highlights the importance of home-based training during the COVID-19 lockdown, off-season, and competitive season in order to minimize the decremental effect of long-term detraining on aerobic capacity, to target improvement in VO2max performance, and reduce injury susceptibility. Considering the relationship between superior aerobic performance and some hematological parameters (hematocrit concentration, erythrocyte, hemoglobin concentration, mean corpuscular hemoglobin concentration) [32–34], it is possibly important that the coaches evaluate their players in terms of hematological parameters at certain time intervals in order to plan and facilitate the return of the players to training, especially after a long-term detraining period.

**Author Contributions:** Conceptualization, A.F.S. and F.M.C.; methodology, F.M.C. and J.A.; formal analysis, F.M.C.; investigation, A.F.S. and J.A.; writing—original draft preparation, A.F.S., F.M.C., G.B., D.Z., R.S., G.G., H.I.C., J.A., F.F. and S.C.; writing—review and editing, A.F.S., F.M.C., G.B., D.Z., R.S., G.G., H.I.C., J.A., F.F. and S.C.; supervision, F.M.C. All authors have read and agreed to the published version of the manuscript.

**Funding:** This research received no external funding.

**Institutional Review Board Statement:** The study was conducted in accordance with the Declaration of Helsinki, and approved by the Institutional Review Board (or Ethics Committee) of Polytechnic Institute of Viana do Castelo. School of Sport and Leisure (code: CTC-ESDL-CE001-2021).

**Informed Consent Statement:** Informed consent was obtained from all subjects involved in the study.

**Conflicts of Interest:** The authors declare no conflict of interest.

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
