# Peer review of "Analysis of the Sustainability of Long-Term Detraining Caused by COVID-19 Lockdown: Impact on the Maximal Aerobic Speed of Under-16 Soccer Players"

_sustainability, doi:10.3390/su14137821_

Round 1

Reviewer 1 Report

I appreciate the authors' effort to look at the extraordinary circumstances of the COVID-19 lockdown in a scientific way. Nevertheless, i think the study is not meaningful enough in terms of content to be published in this journal. 

I would like to base my opinion on three points:

1. Although the lockdowns were unprecedented events, the effect on the training opportunities of athletes is very similar to effects of vacations or injuries. In all cases, the athletes could only train in a limited way. Therefore, I think that the study does not investigate a really new effect. 

2. Only one dependent variable is not enough to get a comprehensive picture of the athletes' performance level. 

3. The study is highly underpowered. Therefore, no valid statements can be made about the lack of effects. 

Reviewer 2 Report

This manuscript examined the effects of de-training during the covid-19 lockdown in youth soccer players. Overall, this manuscript studied a novel topic of interest to the readers. There are several spelling and grammatical errors that should be altered before acceptance. Additionally, some statistics of magnitude of effect would also help in the results section. I will break down these comments in a line by line format below.

Major comments:

In the statistics and results section, the author should calculate some cohen’s d or hedge’s g effect sizes for pre and post measures of change. This can help to show practical outcomes.

Minor comments:

Line 38-40: re-write this sentence. Grammar is incorrect.

Line 45-46: please re-write sentence is hard to read

Line 48-50: should say when training is “resumed” not resume

Line 67-70: please re-write sentence is hard to read

Line 113: Kg should be lower case as kg

Line 113: years of experience overall in football or strength and conditioning experience

Line 115: why was 180 minutes chosen as the cutoff was this the mean of training times

Line 126: define mild temperature in Celsius

Line 128: what was the general warm up? Running, dynamic warm up? What was the time period? Please provide some additional detail.

Line 135: change heigh to “Height”

Line 174-175: is this last line referencing the whole team average? Please define the 2.27% reduction. I am guessing this is the team as a whole .

Line 180-187: change significant to statistically significant

Line 203-205: is this referencing other studies. Need details about athletes, and timeline for this.

Line 217-220: This is a reach. Because you didn’t measure anything outside of MAS you cannot state these other values dropped. I would reword this stating that MAS changed negatively and it may indicate a drop in the other. Without directly measuring them you cannot make this statement.

Line 252-255: this should be re-written as it’s too strong of a statement. This is only one snapshot you cannot say anything is clear from the study.

Line 284-288: This should be re-written typically the best reduction in volume is around 40-60 percent for keeping adaptations around. Could be higher in certain sports. Additionally, frequency should be maintained in trained athletes and in lesser qualified you can remove days. These are pretty clear from the current tapering, peaking, and maintenance literature. 

Reviewer 3 Report

Basically, the submitted manuscript deals with a topic issue. However, there are several major aspects that should be addressed before considering the manuscript for publication:

1.       The authors speak of “long-term detraining”. Please define “long-term” (also see line 258-259: long-term = longer than 4 weeks). A classification (including appropriate references) regarding short-term, (mid-term?) and long-term would be helpful for the reader!

2.       The call for “individualized training” (line 39 and line 308) remains unclear, because the context is missing. Please explain why individualized training during confinement is supposed to be beneficial; otherwise, I suggest renouncing the term individualized.

3.       Line 30: indicate the playing level of the participants, and in this regard:

4.       Line 112: please provide a brief explanation regarding the “national level”. What does this mean? National league, national team? Please briefly elaborate.

5.       Line 83: do you really mean “sedentary”? Please check!

6.       Line 88: Please provide a reference for the definition of “MAS”

7.       Lines 104-105: 6 weeks = 6x7 = 42 days, instead of “52 days” (line 104); However, the difference between the assessment dates is 62 days. Please clarify and explain the difference between the assessment dates and the confinement period – thanks.

8.       The authors discriminated between athletes with low (<180 mins·week-1) and athletes with high (>180 mins·week-1) exercises/PA per week during confinement. First: the allocation to the low or high exercising group remains unclear. Did participants in the high-volume exercise group have to perform >180mins of exercise in every single week, or is the allocation based on a mean value derived from a single questionnaire at the end of the confinement/ at follow-up assessment? => How often was the Physical activity questionnaire applied (weekly/ once in the end)?

9.       Line 116: based on a median value, authors divided participants into two groups. Group 1 N=10 and group 2 N=12. This is not valid, because based on a median value, both groups should have N=11! Please re-check and clarify! In this respect: From the “materials and methods” it appears that some dropouts occurred (see previous point). Therefore, please consider using a figure displaying 1. Number of athletes approached, 2. Number of dropouts (and reasons for drop-out), 3. Number of athletes finally analyzed. Did any Covid-19 infections occur in the sample?

10.   Line 139: please re-check dates (01/09/2020 and 11/02/2020)

11.   Lines 138 & 140: Assessments took place in two different locations. Please state/ indicate that despite these two different locations for the Bronco-test performance, the conditions were identical/ comparable for athletes, and thus the locations had no impact on the performance of the athletes. Otherwise please report in the limitations’ section.

12.   Line 154: please provide a brief explanation for the equation MAS = 1200 / /time in seconds – 20.3). Why is 20.3 subtracted?

13.   Line 155: PA questionnaire: It seems the authors just asked for general physical activity, which includes activities ranging from walking to high-intensity interval training. For sure, the intensity of exercises performed during confinement has a significant impact on the outcome of this study. Also see line 217 (home-based exercises). Is it exercising of general physical activity? Therefore: Did you assess the intensity/ kind exercise performed? If yes: please indicate. If no: please mention in the limitations’ section!

STATISTICS:

14.   The authors indicate absence of normality. Did you try data transform to achieve normalization? Which data were not normally distributed (pre and post values, of difference scores)? Please check if normal distribution can be obtained! If this is the case, then “group x time” interaction can be performed, which has much more statistical power compared to non-parametrical tests!

15.   If normal distribution cannot be obtained through data transformation (e.g., log-transformation) and the authors want to stick to non-parametric testing, then Bonferroni-adjustments must be applied to prevent alpha-error accumulation (e.g., p=0.05/3 comparisons => p<0.0125 (group 1: pre vs. post; group 2: pre vs. post; pre: group 1 vs. group 2; post: group 1 vs. group 2) !

16.   Furthermore, median and interquartile range instead of mean and standard deviation should be reported in the absence of normality!          

17.   Great to see you provide individual change scores!

Limitations

18.   Limitations: The sample size is small, ok. And it is a convenience sample, as indicated in the methods’ section. However, in the limitations’ section the authors should indicate that according to these conditions, no sample size / power calculation was performed. Therefore, any interpretation based on statistical inference from this paper should be cautious! Please add this to the limitations’ section.

19.   Another limitation is that two exercise groups are compared to each other, without really knowing, what was going on within these groups during confinement. In the limitations, it should also be noted that no control group, completely abstaining from exercising, was considered, which would have facilitated the interpretation of performance changes due to the confinement period.

Conclusions:

20.   Line 310: Do you interpret a decrease of 2.3% as “devastating”? Please phrase more cautiously! Same applied to line 314 (“extremely”)!

21.    

Round 2

Reviewer 1 Report

since my review suggested rejection of the manuscript, I will not review the manuscript again. Had I felt that the manuscript could have been improved by revision, my decision would have been "major revision“. The aspects, that lead to my decision cannot be improved by a revision. 

Reviewer 3 Report

Thank you - all points have been addressed.